# A Vaccine Based on Asia1 Shamir of the Foot-and-Mouth Disease Virus Offers Low Levels of Protection to Pigs against Asia1/MOG/05, Circulating in East Asia

**DOI:** 10.3390/v14081726

**Published:** 2022-08-04

**Authors:** Heeyeon Kim, Hwi Won Seo, Ho-Seong Cho, Yeonsu Oh

**Affiliations:** 1Foot and Mouth Disease Division, Animal and Plant Quarantine Agency, Gimcheon 39510, Korea; 2College of Veterinary Medicine and Institute of Veterinary Science, Kangwon National University, Chuncheon 24341, Korea; 3Infectious Disease Research Center, Korea Research Institute of Bioscience and Biotechnology, Daejeon 34141, Korea; 4College of Veterinary Medicine and Bio-Safety Research Institute, Jeonbuk National University, Iksan 54596, Korea

**Keywords:** foot-and-mouth disease, Asia1, FMD vaccine, viral infection

## Abstract

Foot-and-mouth disease (FMD) is one of the most contagious diseases in cloven hoof animals. Vaccination can prevent or control FMD, and vaccine antigens should be matched against circulating viruses. According to phylogenetic analyses, field isolates in this region belonged to genotype V and showed low genetic similarity with the Asia1 Shamir vaccine, the OIE-recommended vaccine strain. In this study, we investigated whether pigs vaccinated with the Asia1 Shamir vaccine could be protected from challenges with the Asia1/MOG/05 virus, one of the genotype V field isolates. Eight pigs were divided into either vaccinated or nonvaccinated control groups. After two vaccinations with Asia1 Shamir, both groups of pigs were challenged with the Asia1/MOG/05 field isolate at 2 weeks after the second vaccination. In the control group, symptoms appeared at 2 days post-infection (dpi). The clinical sign score peaked at 4 dpi, and this coincided with virus shedding through nasal discharge. Neutralizing antibody titers peaked at 17 dpi. In the vaccinated group, clinical signs were delayed compared with the control group, and the highest score was shown at 10 dpi accompanied with virus nasal shedding, which peaked at 11 dpi. Neutralizing antibodies were induced 2 weeks after the second vaccination and peaked at 17 dpi. In conclusion, Asia1 Shamir vaccination in pigs provided partial protection from Asia1/MOG/05 virus infection.

## 1. Introduction

Foot-and-mouth disease (FMD) is one of the most contagious diseases in cloven hoof animals, which is caused by the foot-and-mouth disease virus (FMDV) of the *Aphthovirus* genus and the *Picornaviridae* family [1]. FMDV has a wide host range, an ability to infect in small doses, a rapid rate of replication, a high level of viral excretion, and multiple modes of transmission including being able to be spread by the wind. These features make FMD a difficult and expensive disease to control and eradicate [2]. The FMDV is classified into seven serotypes; each has its own topotypes, and immunity after vaccination or after infection is type-specific [3]. Recovery from infection or protective vaccination with one serotype is known to not protect against subsequent infection with another. Moreover, within a serotype, a wide range of strains may occur; the immunity provided against field strains of the virus that are antigenically different from the vaccine will even reduce the efficacy of the existing vaccines [4]. The Asia1 serotype only occurs in Asia and has only one regional type—ASIA. It is subdivided into genetic group (G)-I to G-IX [5]. After the first outbreak in China in 2005, Asia1-type outbreaks in East Asia occurred continuously, and other FMD outbreaks from the G-V lineage were reported in Mongolia in 2005 as well as in North Korea in 2007 [6]. Outbreaks caused by the FMDV G-VIII lineage, a similar strain that had been circulating in India and Bangladesh, occurred in Myanmar and Nepal in 2017 [7]. A G-VIII (Sindh-08) outbreak occurred in Afghanistan, raising concerns about possible new FMD outbreaks in Asia, where no recent FMD cases had been reported.

The FMDV serotype Asia1 generally infects cattle and sheep, whereas infections in pigs are rarely reported. However, an Asia1-type virus was isolated from naturally infected pigs in 2006 in China [8]. Although the Asia1 type rarely occurs, preventive measures against Asia1 FMD outbreaks are necessary [9]. Asia1 Shamir, the standard international vaccine for the Asia1 serotype, would protect animals against most of the prevalent Asia1 viruses. However, the vaccine was determined to be well-matched against the FMDV G-VIII lineage [7,10]. Despite the current Asia1/MOG/2005 strain harboring, the G-V genotype was reported to not be well-matched with the Asia1 Shamir vaccine [8,10]; there has never been a study examining the efficacy of Asia1 Shamir on the current FMDV in east Asia, especially in pigs. Therefore, the purpose of this study is to evaluate the efficacy of the currently certified vaccine by implementing the vaccination situation in the field and conducting a challenge with an epidemic virus: the Asia1/MOG/05 virus in pigs.

## 2. Materials and Methods

### 2.1. Immunization and Challenge

Virus manipulation was conducted in a biosafety level 3 containment laboratory in accordance with the regulation of the Animal and Plant Quarantine Agency (APQA) in the Republic of Korea. The experimental animals were divided into two groups: four animals in the vaccinated group and four animals in the unvaccinated control group. Pigs in the vaccinated group were double injected with a trivalent commercial vaccine (ARRIAH-VAC^®^ by FGBI ARRIAH, Vladimir, Russia) in field use that contained O primorsky, A Zabaikalsky, and Asia1 Shamir antigens at two-week intervals. All animals were challenged with FMDV Asia1/MOG/05 at a titer of 4 × 10^5^ TCID_50_ on their footpad, respectively, 2 weeks after the second vaccination. Challenged pigs were monitored daily for their body temperature and clinical signs, including vesicle formation, lameness, lethargy, and inappetence, until the end of the experiment. A clinical score was determined by the addition of points as described in a previous study [11]. Briefly, clinical scores were calculated using the following criteria: (i) elevated body temperature of 40 °C (1 point), >40.5 °C (2 points), and >41 °C (3 points); (ii) lameness (1 point) or reluctance to stand (2 points); (iii) the presence of heat and pain (1 point) after palpation of the coronary band (1 point) or not standing on the affected foot (2 point); (iv) hoof and foot vesicles (1 points per foot); and (v) snout, lips, and tongue vesicles (1 point for each affected area).

### 2.2. Sample Collection and Test

Nasal fluids were collected in the period from 0 days post-challenge (dpc) to 17 dpc. The nasal swabs were collected and immediately placed into RNAlater (Viral Transport Kit BD Biosciences, Nottingham, UK), and were stored at −70 °C until required. The viral RNA was identified by quantitative real-time reverse transcription PCR (qRT-PCR), as described elsewhere [12]. The Nextractor 48 system (Genolution, Seoul, Korea) was used for the extraction of the viral RNA, and qRT-PCR was conducted using a one-step FMDV Real-Time RT-PCR MasterMix Kit (Bioneer, Daejeon, Korea) according to the manufacturer’s instructions.

Blood was collected at 2 weeks after the first and second vaccinations, and additionally at 17 and 23 dpc. The sera were heat-inactivated at 56 °C for 30 min. The neutralizing antibody titers in sera were measured using the virus neutralization test (VNT) specified in the Manual of Diagnostic Test and Vaccines for Terrestrial Animals of the OIE. Briefly, following the incubation of the test serum with FMDV 100 TCID_50_ at 37 °C for 1 h, cells were added to the microplate and incubated for three days. The CPE was recorded to determine the titers, which were calculated as the serum dilution necessary to neutralize 100 TCID_50_ of the virus. The FMDVs Asia1 Shamir and Asia1/MOG/05 were used for the VNT. The data are presented as the mean ± standard deviation (SD).

### 2.3. Statistical Analysis

Summary statistics were calculated for all groups to assess the overall quality of the data, including normality. The statistical relationship of clinical sign scores, viral RNA quantification, and VNTs between the vaccinated and control groups were determined with a Student’s *t*-test, followed by the Mann–Whitney U test and Wilcoxon rank-sum test to calculate the post-power of the findings using IBM SPSS Statistics (version 24, New York, NY, USA). Statistical significance was considered as *p* < 0.05.

## 3. Results

### 3.1. Clinical Observation

After the challenge, the clinical sign scores increased in the unvaccinated control group starting from 3 dpc and remained high until 11 dpc compared with the vaccinated group (Figure 1). The pigs of the unvaccinated control group became severely lame and immobilized, and some pigs were observed as having a loss of appetite. From the 14th day of the challenge, regrown epithelium replacing the skin lesion was observed in both of the groups. Body temperature tended to rise, but no difference was observed between the vaccinated and unvaccinated groups (Appendix A).

### 3.2. Detection of Viral RNA

The kinetics of FMDV RNA loads in nasal swabs after the challenge are summarized in Figure 2. Virus shedding started from the third day and continued until 11 dpc. No statistical difference was observed between the vaccinated and unvaccinated groups, including for the peak and decreasing time (Figure 2).

### 3.3. Virus-Neutralizing Antibody Titer (VNT)

The VNTs against Asia1 Shamir and Asia1/MOG/05 are presented in Figure 3. The VNT to Asia1 Shamir started to rise from 14 dpv in the vaccinated group, peaked at 17 dpc, and was maintained until the end of the experiment at 23 dpc. It was statistically significantly higher than that of the unvaccinated group, although the VNT of the unvaccinated group increased after the challenge (Figure 3A). The VNT to Asia1/MOG/05 in the vaccinated group showed a similar pattern to that against Asia1 Shamir (Figure 3B). However, the VNT to Asia1/MOG/05 in the unvaccinated group increased to a similar degree to that of the vaccinated group after the challenge, and was maintained until the end of the experiment, at 23 dpc. A statistical difference between the two groups was observed only before the challenge.

## 4. Discussion

Owing to various mutations in the FMDV, the protective efficacy of the reserved vaccine for use might be reduced. That is, depending on the difference in the antigenic similarity between the vaccine antigen and the circulating virus strain, there could be found to be a gap between the vaccine-induced immunity and the protective potency in field situations. In that sense, it would be necessary to simulate a field situation and properly evaluate the vaccine in possession. Therefore, the aim of this study was to evaluate whether the Asia1 Shamir FMD vaccine, the only official OIE Asia1 vaccine, provides sufficient protection against field epidemics in East Asian pigs.

As the Asia1 G-VIII outbreaks in Nepal and Myanmar were reported in 2017, this virus strain has been confirmed to be well-matched with the Shamir strain and, additionally, the Shamir vaccine was also confirmed to be able to defend against the Asia1 G-VII lineage known as Sindh-08 in Afghanistan [7]. Therefore, Shamir is expected to be able to provide protection at the introduction of the G-VIII and G-VII virus strains. The G-VIII virus outbreaks in Myanmar and Nepal lasted for only a short period and then disappeared, and the G-VII virus was reported in the Middle East. However, the G-V virus was responsible for the 2004–2009 outbreak in East Asia, where FMD has now become endemic [13]. Therefore, in Korea, the burden of the G-V virus outbreak seems to be high. Although Asia1 FMD occurs mainly in cattle and reports of outbreaks in pigs are rare, a widespread occurrence in pigs in the future cannot be excluded in East Asia, including in Korea, where intensive livestock production is carried out [14]. For this reason, Asia1/MOG/20, a G-V genotype isolate, was selected to simulate an infection situation in Korea.

In conclusion, the Aisa1 Shamir FMD vaccine provided partial protection against a G-V genotype isolate, Asia1/MOG/2005. Pigs vaccinated with the Asia1 Shamir vaccine provided delayed and low clinical signs after the challenge compared with unvaccinated pigs. However, fever and virus shedding through nasal routes after the challenge were not prevented via vaccination. VNTs in vaccinated pigs increased after vaccination, although the matching value of Asia1/MOG/2005 against the vaccinated sera was not so high (r1 = 0.25) [15], and remained high after the challenge. However, even in the unvaccinated pigs, an increase in VNTs was observed after the challenge, with no significant difference from the vaccinated group.

It was confirmed that pigs immunized with the Asia1 Shamir vaccine were not provided with complete virological protection. Rather, the results suggested that the vaccine offered partial defense in terms of clinical signs and VNTs. The Asia Shamir vaccine strain is reserved overseas as an antigen bank for future outbreaks in Korea, and is being prepared in many Asian countries. Although the vaccine is found to offer partial protection, the replacement of a vaccine strain for group V requires much in the way of analysis, and although it is changed, it is difficult to guarantee protection against the introduction of viruses, such as group IV (VNT r1 with Shamir: 0.39) [15]. It is believed that countries with an Asia1 risk burden should be sufficiently concerned about this.

## Figures and Tables

**Figure 1 viruses-14-01726-f001:**
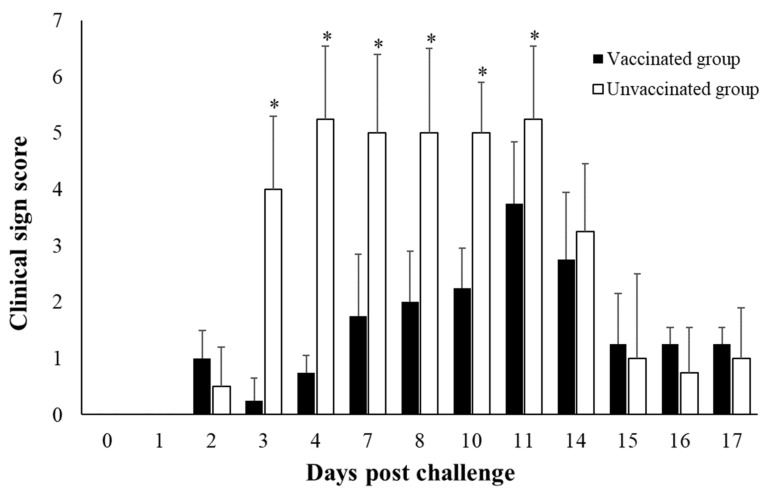
Clinical sign score. *, statistically significance. *p* < 0.05.

**Figure 2 viruses-14-01726-f002:**
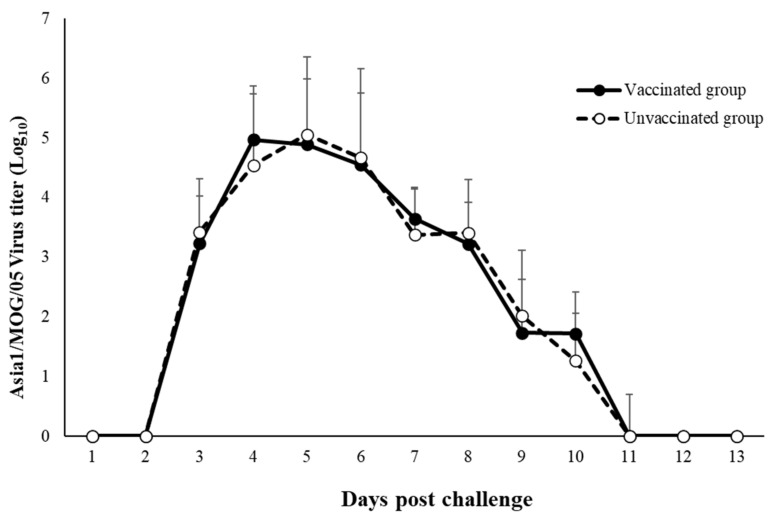
Virus shedding.

**Figure 3 viruses-14-01726-f003:**
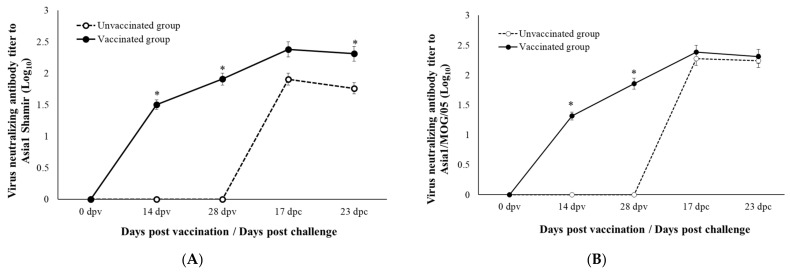
(**A**) VNT to Asia1 Shamir. (**B**) VNT to Asia1_MOG_05. *, statistically significance. *p* < 0.05.

## Data Availability

Not applicable.

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
