# Peer review of "A Vaccine Based on Asia1 Shamir of the Foot-and-Mouth Disease Virus Offers Low Levels of Protection to Pigs against Asia1/MOG/05, Circulating in East Asia"

_viruses, 2022, doi:10.3390/v14081726_

Round 1
Reviewer 1 Report
Figure 3 illustrating VNT data is missing from the manuscript, please have it included. The discussion could be shortened, it needs text editing to improve flow and remove redundancy. I would recommend that in the study design for future vaccine efficacy studies, the period between prime and boosting be extended to 21 days.
Author Response
Thank you for the reviewer’s comment. Figure 3 illustrating VNT data is now included. The discussion part has been through editing to be improved according to the reviewer’s advice. The authors will keep in mind the reviewer’s advice for our future study design.
Reviewer 2 Report
This submission is well presented with good justifications and background. Please consider the below suggestions and comments to improve the presentation and the added value of this study:
1. The sample size for this experiment is limited and the authors are encourage to calculate the post-power of the findings particularly when the outcome is not significant among the two groups;
2. The application of Student t-test to assess the statistical significance between the two groups requires further details such as the response variable (I assume it was the neutralization titer) and the adjustment for the multiple measurement of the same subject. Although a simple statistical test is appropriate for such exploratory experiment, there is a need to conduct the appropriate statistical analysis.
3. It will be great if additional field data of the vaccination with this virus can be shown so that readers can assess the potential significance of the conducted experiment.
4. The limitations of the study should include the power of the findings given the small sample size.
I wish the authors the best to continue this type of studies to complement the actual field control measures of animal diseases.
Author Response
This submission is well presented with good justifications and background. Please consider the below suggestions and comments to improve the presentation and the added value of this study:
- The sample size for this experiment is limited and the authors are encourage to calculate the post-power of the findings particularly when the outcome is not significant among the two groups;
Thank you for the reviewer’s comment. The statistical analysis was already conducted via either parametric or non-parametric methods but was not reflected on the manuscript. The statistical methods we used were revised.
- The application of Student t-test to assess the statistical significance between the two groups requires further details such as the response variable (I assume it was the neutralization titer) and the adjustment for the multiple measurement of the same subject. Although a simple statistical test is appropriate for such exploratory experiment, there is a need to conduct the appropriate statistical analysis.
Thank you for the reviewer’s comment. The description of statistical analysis was revised in the manuscript in detail. Briefly, all parameters to be analyzed between vaccinated and non-vaccinated control groups were described with used statistical methods.
- It will be great if additional field data of the vaccination with this virus can be shown so that readers can assess the potential significance of the conducted experiment.
Thank you for the reviewer’s suggestion. However, it is difficult to add field data because Asia1 virus has never occurred in Korea yet.
- The limitations of the study should include the power of the findings given the small sample size.
Thank you for the reviewer’s suggestion. Although the sample size is small, it is like the actual situation on the farm because field pigs were used instead of SPF pigs. Most of all, since the experimental results of the tested pigs showed very similar patterns without extreme variations, they have the power of the findings in authors’ opinion.
I wish the authors the best to continue this type of studies to complement the actual field control measures of animal diseases.
Thank you so much, we will try to do more studies of animal diseases.

This manuscript is a resubmission of an earlier submission. The following is a list of the peer review reports and author responses from that submission.
Round 1
Reviewer 1 Report
This manuscript provides evidence of poor protection against FMDV ASIA1MOG/05 when ASIA 1 Shamir trivalent vaccine is used. The study design is not optimum, a third group of pigs vaccinated with ASIA 1 Shamir vaccinated and challenged with ASIA1 Shamir should have been included.
The VNT values for ASIA 1/MOG/05 and ASIA 1 Shamir are very similar, thus the importance of confirming the ability of the trivalent vaccine to protect pigs against homologous challenge – thus the importance of the additional test groups.
In addition, clinical scoring should include description of presence or absence of vesicles (feet and snout) would help the reader get a better understanding of what “partial protection looks like. This is fixable if the scoring was conducted properly.
Reviewer 2 Report
The paper describes the lack of efficacy of currently available vaccine for FMDV, due to potentially the difference in the sequences of the neutralizing agn/aby. Although the study may be significant to establish a need for better vaccine, the study design and description is somewhat shallow.
- Fig1 A: The clinical scores and grading criteria were not defined.
- Fig 2 : Viral RNA was measured but the graph is shown as virus titers.
- Fig 2 showed no protection, not partial protection.
- Fig 3 and 4 : Is the time scale based on post-vaccination or post-challenge?